# BLOX: Macro Neural Architecture Search Benchmark and Algorithms

**Thomas Chau[1*], Łukasz Dudziak[1*], Hongkai Wen[1,3]**
**Nicholas D. Lane[1,2], Mohamed S. Abdelfattah[4]**

[1] Samsung AI Center, Cambridge, UK
[2] University of Cambridge, UK     [3] University of Warwick, UK
[4] Cornell University, USA
[*] *Indicates equal contributions*

{thomas.chau, l.dudziak, hongkai.wen, nic.lane}@samsung.com
mohamed@cornell.edu

## Abstract

Neural architecture search (NAS) has been successfully used to design numerous high-performance neural networks. However, NAS is typically compute-intensive, so most existing approaches restrict the search to decide the operations and topological structure of a single block only, then the same block is stacked repeatedly to form an end-to-end model. Although such an approach reduces the size of search space, recent studies show that a macro search space, which allows blocks in a model to be different, can lead to better performance. To provide a systematic study of the performance of NAS algorithms on a macro search space, we release Blox – a benchmark that consists of 91k unique models trained on the CIFAR-100 dataset. The dataset also includes runtime measurements of all the models on a diverse set of hardware platforms. We perform extensive experiments to compare existing algorithms that are well studied on cell-based search spaces, with the emerging blockwise approaches that aim to make NAS scalable to much larger macro search spaces. The Blox benchmark and code are available at https://github.com/SamsungLabs/blox.

## 1   Introduction

Deep neural network (DNN) performance is closely related to its architecture topology and hyper-parameters as demonstrated through the progression of image classification CNNs in recent years: AlexNet [1], Inception [2], MobileNets [3] and EfficientNets [4, 5]. Increasingly, automated methods are used to design DNN architectures to avoid intuition-based manual design. The field of neural architecture search (NAS) continues to offer a large number of methods including sample-based NAS [6, 7], differentiable NAS [8], training-free NAS [9] and blockwise NAS [10, 11, 12]. Within this realm of NAS for DNN design there are two important design problems which are still mostly manual. First, how do we design the NAS *search space*, which defines the set of DNN architectures from which a NAS algorithm can select. Second, how do we select or design a suitable search method for a given NAS search space. In this work, we attempt to address both problems through a focused analysis of *macro* NAS algorithms within a new NAS search space called Blox.

**NAS search spaces.** A well-defined search space is crucial for NAS. However, the literature has mostly focused on cell-based designs in which the NAS algorithm only searches for operations and connections of a cell that is repeatedly stacked within a predefined skeleton [8, 13, 14, 15, 16, 17, 18]. These approaches prohibit layer diversity which can help to achieve both high accuracy and low

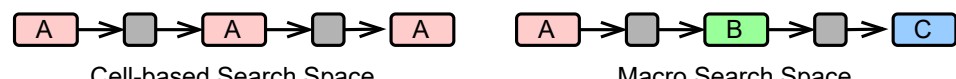

Cell-based Search Space        Macro Search Space

Figure 1: A macro search space allows each block to have a different architecture whereas a cell-based search space repeats the same cell/block throughout the DNN.

latency [19]. An alternative to cell-based NAS, known as macro NAS, enables the individual search for each block in a DNN as shown in Figure 1. In other words, macro NAS allows different stages of a model to have different structures. Though promising, macro NAS is exorbitantly expensive because the search space size grows exponentially with the number of blocks. We present the first large-scale benchmark and study of a macro search space to shed some light on how to perform NAS in this challenging setting.

**NAS benchmarks.** To facilitate a fair comparison of NAS algorithms, standardized benchmarks have been created such as NAS-Bench-101/201/1shot1/NLP/ASR [13, 14, 20, 21, 22]. While these benchmarks span multiple application domains, they all use cell-based search spaces thus limiting the analysis of NAS algorithms to this setting only. More recently, NAS-Bench-Macro [23] proposed a macro search space with 8 stages; however, each stage only has three block options making the overall search space quite small ($3^8 = 6,561$ DNNs) and not diverse. To address this, we have developed Blox – a much larger macro NAS benchmark that focuses on block diversity, with 45 unique block options and three stages ($45^3 = 91,125$ DNNs). This enables the empirical analysis of NAS algorithms on macro search spaces and will thus inform the design of efficient macro search algorithms. Table 1 summarizes Blox and other recent NAS benchmarks.

**Macro NAS algorithms.** Any search algorithm can operate on a macro search space; however, very few will be efficient because of the large search space size. To cope with the complexity of macro search, a new class of *blockwise* search algorithms are being developed that perform local search within each stage before using that local information to construct an end-to-end model. Blockwise search algorithms is a family of NAS algorithms designed to work well for macro NAS problems. This divide-and-conquer approach has the potential to speed up macro NAS at the expense of using inexact heuristics to predict the performance of each block. DNA [10], DONNA [11] and LANA [12] are three recent and notable works in this area, showing state-of-the-art accuracy-latency tradeoffs on very large macro search spaces. In this work, we aim to analyze the different components of these blockwise NAS algorithms to understand, compare and build upon the existing approaches.

We enumerate our contributions below:

1. **Macro search space and benchmark for NAS.** We release Blox, a NAS benchmark for CNNs on a macro search space, trained on the CIFAR-100 dataset [24], with latency measurements from multiple hardware devices.
2. **Analysis of blockwise NAS.** We systematically evaluate the performance of different NAS algorithms on Blox, with a particular focus on emerging blockwise search algorithms, for which we include a detailed analysis of the efficacy of (a) block signatures, (b) accuracy predictors, and (c) training methodologies.

## 2 Blox: Macro Search Space

Blox is a macro search space for CNNs on image classification task. The search space is designed to be compatible with all NAS methods, including differentiable architecture search [8].

### 2.1 Search space

We opt for a simple search space definition that mimics many recent CNN architectures [3, 4, 5] and NAS search spaces [8, 14]. Figure 2 shows an overview of the Blox search space. The network architecture consists of three stages, each containing a searchable block and a fixed reduction block. The searchable block can be expressed as a differentiable supernet as shown in the figure (block architecture), and is allowed to be different for each stage to construct the macro search space. We designed the block architecture to allow for interesting and diverse connectivity between operations as

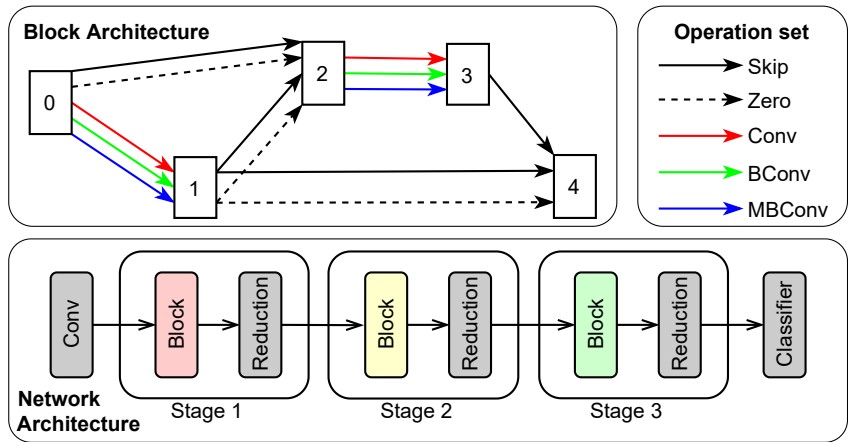

Figure 2: Overview of the Blox macro search space.

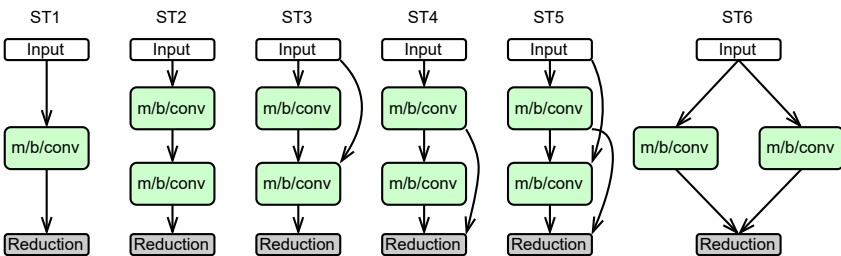

Figure 3: Example block architectures from Blox showing diverse connectivities. Conv: VGG-style [26] 3x3 convolutions. BConv: Resnet-style [25] bottleneck with 5x5 depthwise-separable convolutions. MBConv: EfficientnetV2 fused-inverted residual convolution [5] including squeeze and excitation operation [27].

exemplified in Figure 3. We make sure to include common structures such as residual connections [25] and inverted bottleneck blocks [5] that are relevant for the state-of-the-art CNNs. In total, there are 45 unique blocks, making the size of the Blox search space $45^3 = 91,125$. Additionally, we selected operations from the relevant state-of-the-art DNNs [3, 5, 25, 26], and controlled their repetition factor to roughly balance FLOPs and parameters across the different blocks (more details in the supplementary material).

## 2.2 Training details

Throughout the paper we consider models from our Blox search space in 3 different training scenarios: *1) Normal* setting is when a model is trained in a standard way, without any other model participating in the process. Information about the performance of all 91,125 models when trained normally comes pre-computed with our benchmark; *2) Distillation* refers to a setting in which individual candidate blocks are distilled independently to mimic analogous blocks from a normally-trained teacher model $T$ – this process is described in more details in section 3; *3) Fine-tuning*, which follows *Distillation*, is a process when blocks that were distilled independently are used to form an end-to-end model $M$. $M$ is then trained using the standard knowledge distillation approach with the same teacher $T$ which was used to distill blocks. For the information about hyperparameters used for each of the three settings and what is included with the benchmark, please refer to the supplementary material.

Blox currently provides tabular results of training-from-scratch to enable systematic study of conventional NAS algorithms on emerging macro search spaces. Such results are also beneficial for studying blockwise algorithms (even though it does not directly enable their fast evaluation) because it allows better control of parameters of experiments (e.g. choosing "good teacher vs. bad teacher"), and enables comparison of the accuracy of the same models trained using different approaches.

Table 1: Comparison to other NAS benchmarks.

| | # models | Type | Operations |
|---|---|---|---|
| NAS-Bench-101 [13] | 423k | | conv3x3, conv1x1, maxpool3x3 |
| NAS-Bench-201 [14] | 15,625 | | zeroize, skip connection, conv1x1, conv3x3, avgpool3x3 |
| NAS-Bench-1shot1 [20] | 363k | cell-based | conv3x3, conv1x1, maxpool3x3 |
| NAS-Bench-NLP [21] | 14,322 | | linear, element wise, activations (Tanh, Sigmoid, LeakyReLU) |
| NAS-Bench-ASR [22] | 8,242 | | linear, conv1x5, conv1x5 dilation2, conv1x7, conv1x7 dilation2, zeroize |
| NAS-Bench-Macro [28] | 6,561 | macro | identity, MB3_K3, MB6_K6 |
| Blox | 91,125 | | conv, bconv, mbconv |

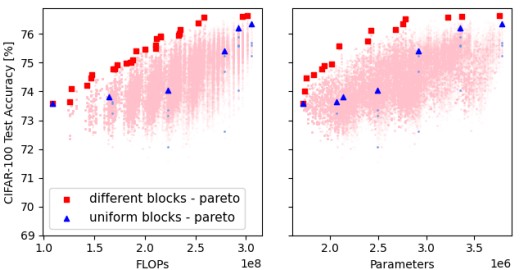
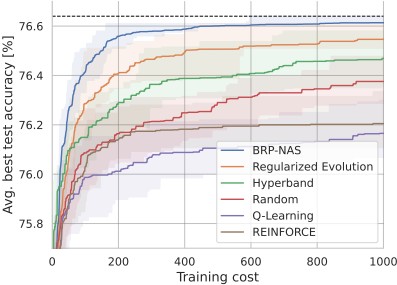

Figure 4: Accuracy v.s. FLOPs and parameters for all models in the Blox space. The Pareto-frontier of models with different blocks dominates that of models with repeated "uniform" blocks – only macro NAS can discover the superior models.

Figure 5: Comparison of conventional NAS search algorithms on Blox. For the details about each algorithm, please see Appendix C.2.

## 2.3 Differences to other NAS benchmarks

We summarize characteristics of Blox and other recent NAS benchmarks in Table 1. In order to highlight both promises and challenges of macro NAS versus cell-based NAS, Figure 4 shows the accuracy of cell-based models consisting of uniform blocks (blue), and macro models consisting of different blocks (red), from our Blox search space when plotted against their number of FLOPs or parameters. Pareto-optimal points are additionally emphasized with markers. There are two highlights. *1)* The Pareto-frontier of models with different blocks clearly dominates that of models with uniform blocks. It indicates that a macro search space contains higher performing models than a cell-based search space thus motivating our benchmark and study; *2)* There are many more models with different blocks than the models with uniform blocks. The macro search space is much larger, posing a challenge to the searching algorithms. Figure 4 shows the trade-off between achievable results and the amount of configurations available. Every cell-based search space can be turned into a much larger macro search space, which leads to a much higher exploration cost, and the achievable accuracy would likely improve.

Comparing to NAS-Bench-Macro, another published macro search space, Blox considers a larger number of diverse replacements. In terms of individual linear operations (e.g. a single convolution), the shallowest block out of the 45 candidates in Blox contains only 4 layers while the deepest block has 36 layers. This means that the depth of the whole network can range from 12 to 108 layers (excluding fixed parts). The design of Blox follows a complementary approach which uses a lower granularity of blocks with more diverse replacements, while NAS-Bench-Macro focuses on the opposite direction with higher granularity of blocks which results in lower diversity of candidates (e.g. NAS-Bench-Macro contains a single operation without any choices regarding connectivity, while Blox uses 2 operations per searchable stage thus introducing another degree of freedom related to the connections between them). Having NAS benchmarks that explore different design choices increases our opportunities to study NAS algorithms in different situations and better understand their behaviour.

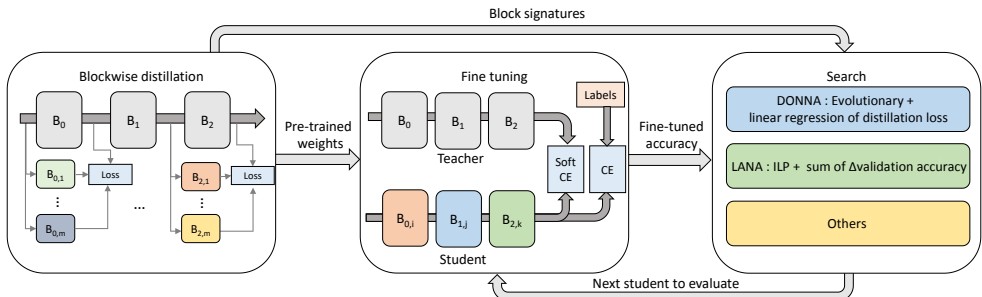

Figure 6: Blockwise NAS – (1) *Blockwise distillation* is performed to obtain the signature of each candidate blocks. (2) *Fine-tuning* initializes the blocks of student model with weights obtained in distillation. Then the student model is trained with knowledge distilled from the teacher. (3) *Search* is conducted using different NAS algorithms to find the best model after fine-tuning.

## 3  Experiments: Characterizing Blockwise NAS on Blox

In order to alleviate the challenges associated with macro NAS, *blockwise* algorithms have been proposed recently and showed promising results in optimising state-of-the-art models on large-scale image classification [10, 11]. Figure 6 shows an overview of blockwise NAS methods: *1) Blockwise distillation* divides a pre-trained reference model (teacher) into sequential blocks that are later distilled into their possible replacements independently from each other. The process of blockwise distillation produces a *library* of pre-trained replacement blocks together with their *signatures*, such as distillation loss or drop in the teacher's accuracy when a student block is swapped-in; *2) Search* uses these signatures to guide an algorithm to find well-performing models built by stacking a number of blocks from the block library; *3) Fine-tuning* is a process when blocks of student model are initialized with weights obtained in distillation, then the model is trained with knowledge distilled from the teacher.

Although outstanding results were demonstrated, blockwise NAS algorithms have not been thoroughly evaluated yet, presumably due to their exceptionally challenging setting. To the best of our knowledge, their performance has not been evaluated in a common setting, nor compared to standard NAS methods, and very few of their design choices and assumptions have been adequately investigated. In this section, we attempt to fill these gaps with the help of our Blox search space and benchmark.

We begin by establishing a baseline by running conventional NAS algorithms that can be found in the literature in the simplest setting when each model is trained normally – results are shown in Figure 5. The relative efficiency of our search algorithms matches what is found in the literature. Binary-relation predictor-based NAS (BRP-NAS) [29] performs best, followed by evolutionary search [7] then other methods [30, 31, 32]. Other than providing these measurements to accompany our benchmark, we aim to compare to the two most recent blockwise NAS algorithms in the remainder of the paper – **DONNA** [11] employs a block-level knowledge distillation technique. Each block's distillation loss is treated as its signature. To perform search, an accuracy predictor (linear regression model) is trained by sampling and fine-tuning random architectures once all blocks are distilled. The predictor takes the block signatures as input and predicts the performance of models. This accuracy predictor guides an evolutionary search over the search space to find models that satisfy performance constraints. **LANA** [12] also uses blockwise distillation to train a library of blocks. The block signature is the change of teacher's validation accuracy after a block is swapped with the candidate block. Then an integer optimization problem, which minimize the sum of block signature, is used to select efficient models.

### 3.1  Fine-tuning versus normal training

We ask questions related to the performance of models when they are fine-tuned in the blockwise setting compared to that when trained normally, with special attention to the implications for NAS.

**Q1: Does distillation help us achieve better performance compared to normal training?** Distillation from a teacher model is a central part of both DONNA and LANA, at the same time there is plenty of evidence in the existing literature suggesting that distillation helps a model achieve better

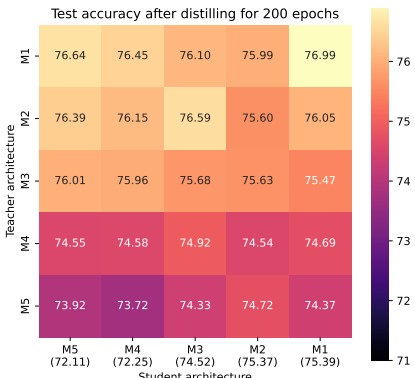

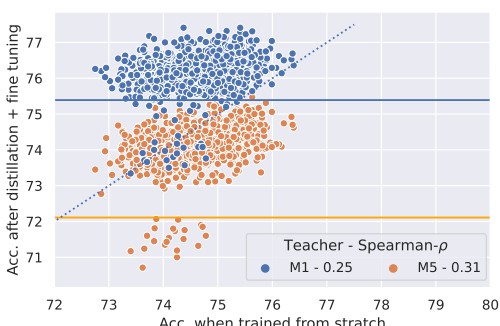

Figure 7: Mutual effect of the student and teacher architecture on the distillation outcome. Each cell at position $(x, y)$ contains information about accuracy when model M$x$ is blockwise-distilled and then fine-tuned from a normally-trained model M$y$. Performance of each model when trained normally is included in the X-axis' tick labels for reference.

Figure 8: Spearman correlation between fine-tuned accuracy and training-from-scratch accuracy. For fine-tuning, all models are trained for 200 epochs, either using the good teacher (M1) or the bad teacher (M5). Solid lines mark performance of each teacher and dashed line marks $y = x$ diagonal.

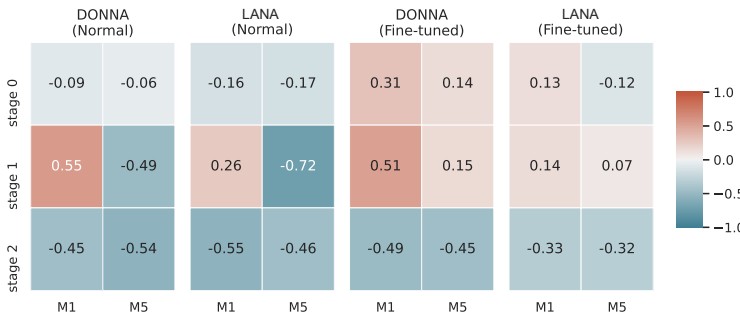

Figure 9: Spearman correlation between block signatures and oracle ranking for 1000 random models. The oracle rank of a block is defined by the best normal / fine-tuned accuracy of a model containing that block. M1 and M5 are used as the teachers.

results, often exceeding even the teacher's performance. However, it is important to note that the setting in those works is usually very different from NAS. Specifically, distillation is conventionally used in situations when the teacher model is known to perform better than the student (e.g., it is significantly larger) – in general, this important assumption might not hold in a NAS setting when we sample models from a search space without knowing if they are better or worse than our teacher. In order to investigate the expected outcome of fine-tuning in different scenarios, we select 5 different architectures from our search space: 2 from the top performing ones, 1 average, and 2 bad ones; we refer to them as M1-M5, where M1 is the most accurate and M5 is the least accurate among them. We then run blockwise distillation for 10 epochs and fine-tuning for 200 epochs (to match normal training) for each of the 25 possible $(student, teacher)$ pairs. From the results in Figure 7, we can see that in all cases **student model is able to improve upon its teacher, delivering on the promise of blockwise distillation from the existing works.** However, we can also see that **compared to the accuracy achievable when a student is trained normally, fine-tuning does not always result in improvement.** Specifically, when a bad teacher is used, accuracy of the models that otherwise tend to achieve good performance is suppressed – this can be seen in the lower-right corner of Figure 7.

**Q2: Does fine-tuning accuracy correlate to training-from-scratch accuracy?** Figure 7 includes one more significant observation for NAS – even within our small sample of 5 models relative ranking of models after fine-tuning is different from when the models are trained normally. For example, when M1 is used as a teacher we can see that the second best model turns out to be M5, which normally is

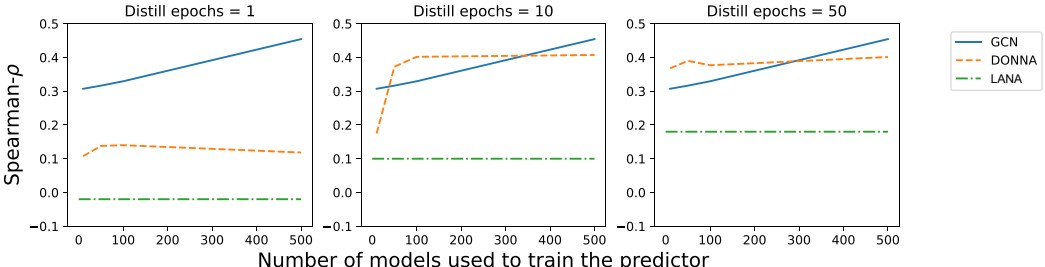

Figure 10: Comparison of different predictors on estimating end-to-end model accuracy (after distillation and fine-tuning). Y-axis shows the Spearman correlation between the predicted and actual accuracy. In this experiment, 1000 models are randomly sampled from Blox. The number of models used to train the predictors are indicated in the x-axis, and the rest of the models are used for testing.

the worst. This suggests that **models exhibit vastly different performance when distilled compared to normal training.** To further investigate this behaviour, we scale up our analysis to 1000 random student networks which are distilled with M1 and M5, then we correlate their training-from-scratch accuracy to fine-tuning accuracy. Results are presented in Figure 8. We can see that indeed on the larger sample correlation remains rather weak for both teachers, with Spearman-$\rho$ of $0.25$ for M1 and $0.31$ for M5. At the same time, Figure 8 further supports our observation that fine-tuning is only beneficial if a good teacher is used – even though most of the students are able to significantly improve upon the M5 teacher, most of them do not improve upon their own training-from-scratch accuracy; this is not the case for a good teacher though. Poor correlation between fine-tuning and normal training suggests that in general we should not perform NAS by simply searching for a good model using standard training and then rely on distillation to boost its performance – instead, **we can achieve better results if we directly search for a model that performs well when distilled.**

### 3.2 Searching for good students efficiently

In the previous subsection we showed that blockwise distillation can be helpful in improving accuracy of models, and it is important to identify good students under distillation settings as fast as possible in order to minimize searching cost. We also highlighted that blockwise methods utilise a divide-and-conquer approach where signatures of different blocks are used to guide the search. We therefore ask the following questions related to block signatures and their usage in NAS.

**Q3: How well do block signatures identify good blocks?** We compare different block scoring methods by measuring their correlation with an *oracle ranking*. The oracle ranking is computed by answering *"if this block is selected at this stage, what is the best accuracy we can get?"* for each candidate block, and the blocks are then sorted accordingly. Figure 9 compares the two block signatures, distillation loss (DONNA) and change of validation accuracy (LANA), in their ability to identify good blocks with approximated oracle ranking, when distilled from a good (M1) and a bad (M5) teacher. It is the first time that the efficacy of DONNA and LANA block signatures are quantified, and surprisingly, **they are not consistently indicative of block performance.** In particular, the correlation of the last stage (stage 2) is much worse than the earlier stages.

**Q4: Can we still use signatures to predict end-to-end performance?** Even though signatures are not good indicators when it comes to identifying if individual blocks would lead to the best possible model on their own, it is still possible that they can be used in a smart way to estimate end-to-end performance. In DONNA, a linear regression model with second-order terms is used to predict end-to-end accuracy, using block signatures as features and accuracy as targets. In LANA, a simple sum of signatures is used as a proxy to approximate the non-linear objective to minimize the loss function. At the same time, there are predictors that utilize graph structure rather than block signatures and deliver promising results in other settings. For example, BRP-NAS [29] uses a graph convolutional network (GCN) to capture graph topology and predict performance of a model. We compare these different prediction-based approaches in estimating end-to-end model accuracy.

Figure 10 shows comparison of different predictors used to estimate end-to-end model accuracy after distillation and fine-tuning. There are 3 findings: *1)* **Distillation loss + linear regression (DONNA)**

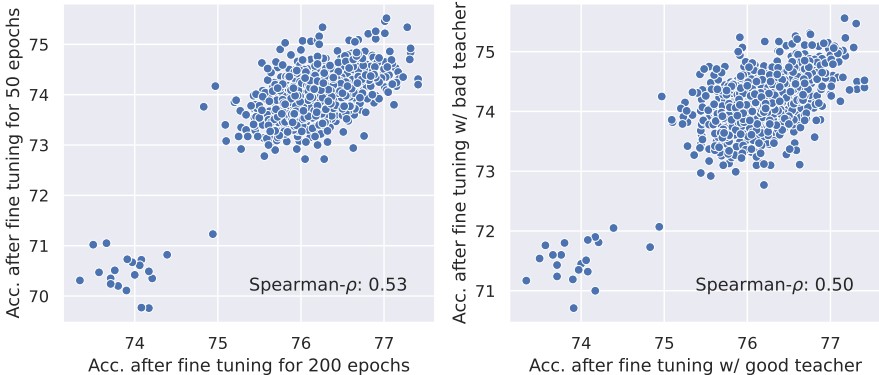

Figure 11: (Left) Spearman correlation between fine-tuned accuracy for 50 epochs and 200 epochs, using the good teacher (M1). It indicates that high performing models can be identified by fine-tuning for fewer number of epochs. (Right) Spearman correlation between fine-tuned (for 200 epochs) accuracy using the good teacher (M1) and bad teacher (M5). It indicates that high performing models can be identified even by using the bad teacher.

is better than change of validation accuracy + simple summation (LANA). *2)* **Signatures of blocks that were distilled for more epochs tend to produce better predictors. 3) The GCN predictor, which does not require any distillation signatures, can outperform DONNA. However, DONNA works better with a small number of training points, provided that the blockwise distillation was performed with 10 or more epochs.**

**Q5: Do we have to fine-tune for 200 epochs?**

It is possible that reduced fine-tuning (e.g. for 50 epochs) can identify models that are as good as those found by the full searches (e.g. for 200 epochs). To investigate this, Figure 11 (left) shows the correlation of fine-tuning for 50 and 200 epochs – the results suggest that **it should be possible to still identify good models without full fine-tuning.** To confirm this hypothesis, in Figure 12, we first search by fine-tuning the student models for 10 epochs (FT10, which has 400 models trained when the training cost reaches 30 as indicated by the gray line). Then we rank the models searched and retrain them for 200 epochs using the same teacher. The big improvement seen in the blue curves indicated that the models searched are good. After full training they outperform the models searched by fine-tuning the student models for 200 epochs (FT200, which only has 20 models searched when the training cost reaches 30). This aligns with the results in Figure 11 (left) that models trained with 50 epochs and 200 epochs are highly correlated.

**Q6: Can we search for good models without prior knowledge of a good teacher?** Figure 11 (right) shows the correlation of fine-tuning by different teachers. We can see that the accuracy of models fine-tuned by a good teacher is highly correlated with that of a bad teacher. Although we know that model selection is robust to the choice of teacher, it is not usually the case that we have prior knowledge of a good teacher. In such case, we propose to find better teachers iteratively: 1) Start NAS and fine-tune models using any teacher. 2) Stop the search and obtain a list of candidate models. 3) Train the candidate models from scratch and pick the best model as the teacher. 4) Fine-tune the candidate models using the teacher selected, and pick the best model as the next teacher. 5) Repeat step 4 until we see no further improvement. As we can see from the blue curve in Figure 13, **the iterative approach has significantly improved the model accuracy without knowing a good teacher in advance.**

### 3.3 Comparison of different NAS methods

Table 2 quantifies the performance of different methods on the Blox search space. We measure two things – accuracy after reaching a fixed cost of 40, and cost required to achieve an accuracy of 76.6 (which is roughly the accuracy of the best model in our search space when trained normally). For conventional NAS, we highlight *regularized evolution* [7] and BRP-NAS [29] which have the best results among the others, and *DART-PT* [33] which is a well-known differentiable NAS method.

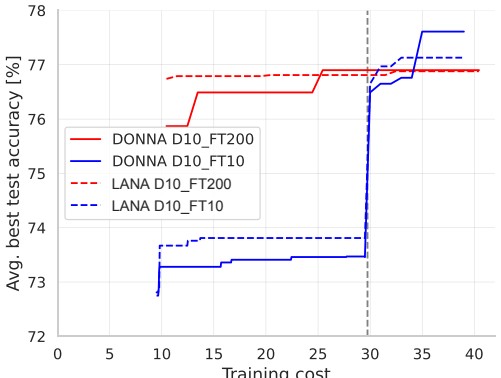
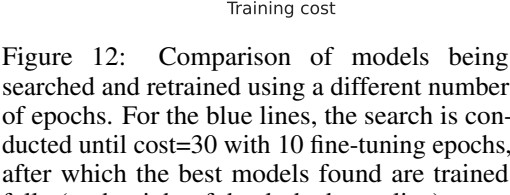

Figure 12: Comparison of models being searched and retrained using a different number of epochs. For the blue lines, the search is conducted until cost=30 with 10 fine-tuning epochs, after which the best models found are trained fully (to the right of the dashed gray line).

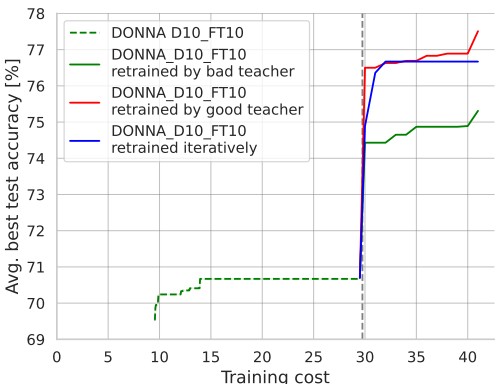

Figure 13: Models being searched using a bad teacher (M5), and retrained using different schemes. Firstly the search is conducted till cost=30 as indicated by the gray line. Then the models found during search are ranked and retrain for 200 epochs using 3 different schemes – (1) the same bad teacher (green curve), (2) the good teacher (red curve), (3) iterative approach (blue).

Table 2: Performance of different NAS methods on Blox search space. For blockwise-NAS methods, the blocks are distilled for 10 epochs and then the models are fine-tuned using a *FTα* setting, e.g. *FT10* means the models are fine-tuned for 10 epochs in the search process. In the cases that retraining is performed (which starts once the cost reaches 30), the searched models are ranked and retrained until the cost or accuracy target is reached, e.g. *FT10 + FT200* means the models are retrained for 200 epochs. If a different teacher is used for retraining, it is denoted, e.g. *FT10 + FT200 iter.* means the teacher used in the retraining process is selected by the iterative strategy. The middle column shows the accuracy achieved after reaching a fixed cost of 40, and the last column shows the cost required to achieve an accuracy of 76.6. ✗ means the target accuracy cannot be achieved with reasonable cost. † DARTS-PT is a differentiable NAS method that the cost of 40 does not apply.

| Method | Acc. @ cost=40 | Cost @ acc.=76.6 |
|---|---|---|
| Conventional NAS | | |
| BRP-NAS | 76.40 | 400 |
| Reg. Evolution | 76.10 | ✗ |
| DARTS-PT | 74.52† | ✗ |

| Method | Acc. @ cost=40 | Cost @ acc.=76.6 |
|---|---|---|
| Blockwise NAS *assuming good teacher (M1)* | | |
| FT200 | 76.90 | 25 |
| FT10 | 73.47 | ✗ |
| FT10 + FT200 | 77.66 | 30 |
| Blockwise NAS *assuming bad teacher (M5)* | | |
| FT10 | 70.67 | ✗ |
| FT10 + FT200 | 74.90 | ✗ |
| FT10 + FT200 iter. | 76.67 | 30 |
| FT10 + FT200 M1 | 76.88 | 30 |

We can see a few things: *1)* Conventional NAS achieves worse results than standard blockwise (FT200) when a good teacher is used; *2)* We can improve blockwise NAS by utilising reduced fine-tuning proxy followed by full fine-tuning (FT10 + FT200), which is our contribution stemming from questions Q 1-5; *3)* However, when a bad teacher is used (FT10 + FT200 at the bottom part), blockwise NAS actually falls short to its conventional counterpart – the results can be improved by our proposed simple iterative strategy (FT10 + FT200 iter., Q 6), which allows us to again dominate conventional NAS. In fact, iterative strategy is almost as good as using the good teacher in the second phase of the search (FT10 + FT200 M1).

Overall, our results again showcase the dominant role of the final fine-tuning and, more broadly, quality of training in blockwise NAS. We include more detailed discussion about interpretation of some of our results, as well as their limitations, in the supplementary material.

## 4   Conclusion

In this work, we presented Blox – a macro NAS search space and benchmark designed to provide a challenging setting for NAS. With its help, we perform a thorough analysis of the emerging blockwise NAS algorithms and compare them to each other and the conventional NAS methods that can be found in the literature. Our results include a quantitative analysis of the efficacy of block signatures and accuracy predictors, furthermore, we discover that the training methodology, especially the teacher model architecture during distillation, plays a bigger role than student model architecture in finding a good student model. Our findings are somewhat unexpected and only made possible by the availability of Blox, for which we hope to see many more interesting studies by the research community.

**Acknowledgments and disclosure of funding**

This work was done as a part of the authors' jobs at the Samsung AI Center, and was supported by the National AI Strategy Fund at the Alan Turing Institute. We thank the reviewers of the NeurIPS Datasets and Benchmarks Track 2022 for their comments and suggestions that helped improve the paper.

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
