# OpenReview forum: "BLOX: Macro Neural Architecture Search Benchmark and Algorithms"
_NeurIPS.cc/2022/Track/Datasets_and_Benchmarks — NeurIPS 2022 Datasets and Benchmarks _

### Official Review · Reviewer_dMTQ · 2022-07-19
**An interesting benchmark on macro NAS algorithms**

**Rating:** 7
**Confidence:** 3
**Correctness:** The claims made in the paper are main…
**Clarity:** Well written.

**Strengths:**

This is a timely work on the important research topic (NAS), and the paper is well written and organized. Authors conduced comprehensive experiments to characterize the blockwise NAS.

**Weaknesses:**

I don't think this paper has any obvious shortcomings, but it can be improved in some small parts:
(1) The authors should clarify the reason or insight of your model architecture search space. Why do you design this way and what are the benefits?
(2) I appreciate the comprehensive experiments conducted against Q1-Q6, though they are interesting, but a little bit hard to follow, for example, why do you ask these questions and how are they connected, and please also highlight the concrete answer (bold) for each question.


**Additional Feedback:**

Nope.

**Documentation:**

Good documentation

**Ethics:**

No ethical concern

**Relation To Prior Work:**

Yes

**Summary And Contributions:**

Authors release a macro block based NAS benchmark called BLOX that includes 91k models trained on cifar100 dataset. This benchmark includes the performance and runtime measurements on different hardware platforms. Authors also perform comprehensive comparison between the block-wise NAS and cell-based NAS.

---

> ### Author Response · Authors · 2022-08-22
> **Thank you for the comments**
>
> We sincerely thank the reviewer for the valuable comments.
>
> * Regarding the insight of the macro search space, please see our detailed answer in **Section S4**.
> * Regarding the rationale behind the questions and the conclusion we have drawn, please refer to **Section S6**.
>
> We hope the responses above will resolve your concerns. If you have any more questions, please let us know and we would be happy to continue the discussion.

---

> > ### Comment · Reviewer_dMTQ · 2022-08-29
> > **Thanks for the response.**
> >
> > Thanks for the response. I‘ll remain my origin score.

---

### Official Review · Reviewer_pU8w · 2022-07-20
**Interesting NAS study but lacking as a benchmark**

**Rating:** 6
**Confidence:** 4
**Correctness:** Claims are correct except for those d…
**Clarity:** Yes.

**Strengths:**

- The benchmark precomputes 95K architectures that accelerates architecture evaluation for future NAS research.
- Experiments make a strong case for fine-tuning, which is effective with a good teacher.
- Block signatures show promise as a performance predictor.
- Runtime under different hardware settings is measured and provided.
- Code is open-source in anonymized repository, alongside good benchmark documentation in the appendix.

**Weaknesses:**

- Benchmark is only evaluated on CIFAR-100. This choice should be justified. Typically CIFAR-10 is the common denominator for NAS benchmark datasets.
- Another advantage of multiple dataset is the possibility of studying distilling the teacher on a new dataset, which could demonstrate this methodology’s usefulness. This study is not enabled in the current setting.
- I remain doubtful of the search space’s quality, given it’s best accuracy of 76.6% on CIFAR-100. P-DARTS reports more than 80% accuracy for most methods in DARTS-like search space on this dataset (see [this link](https://arxiv.org/pdf/1904.12760v1.pdf)). Therefore, the superiority of macro search spaces is not clearly established.
- Only a limited number of NAS methods are evaluated on this tabular benchmark, which excludes weight-sharing (one-shot) methods that are state-of-the-art.
- Additional seeds for the experiments are forthcoming.
- The code repository does not seem to contain scripts for experiments in Sec 3.

**Additional Feedback:**

I'm looking forward to engage with the authors on the issues mentioned.

**Documentation:**

Full details are provided.

**Ethics:**

No ethical concerns.

**Relation To Prior Work:**

Yes.

**Summary And Contributions:**

BLOX introduces a new macro NAS search space and explores NAS block algorithms through research questions regarding distillation. The contribution consists of a precomputed benchmark and insights from controlled experiments.

---

> ### Author Response · Authors · 2022-08-22
> **Thank you for the comments**
>
> Thank you for the insightful feedback.
>
> * Benchmark is only evaluated on CIFAR-100
>   * Thanks again for the comments. Please refer to **Section S5** for our justification.
>
> * Possibility of studying and distilling the teacher on a new dataset is not enabled in the current setting
>   * Blox currently provides tabular results of training-from-scratch to enable systematic study of conventional NAS algorithms on emerging macro search spaces. Such results are also beneficial for studying blockwise algorithms. We have a more detailed discussion in **Section S7**.
>
> * Quality of search space
>   * Please refer to **Section S4**.
>
> * Only a limited number of NAS methods are evaluated
>   * Please refer to **Section S3** on how we have used Blox to evaluate various common NAS methods (BRP-NAS, Regularised evolution, Hyperband, Random, Q-Learning and REINFORCE).
>   * Regarding one-shot methods, we have run DARTS-PT on Blox and have **added the results to Section 3.3 and Table 2**.
>
> * Additional seeds for the experiments are forthcoming
>   * We will include additional seeds for the dataset in the future, at the moment we are bound by the time and compute resources required to train the models multiple times.
>
> * The code repository does not seem to contain scripts for experiments in Sec 3.
>   * We have **provided the scripts to train and generate the Blox dataset**. Regarding the scripts for experiments, they are related to the NAS algorithms rather than the proposed search space, therefore, it is implemented in a separate code repository which we are working towards open-sourcing.
>
> We hope the responses above will resolve your concerns, and would consider increasing the score. If you have any more questions, please let us know and we would be happy to continue the discussion.

---

> > ### Comment · Reviewer_pU8w · 2022-08-29
> > **Response**
> >
> > I am happy to see that more NAS evaluations are being added to this work. For the time being, I think it is acceptable to only consider one dataset. I will increase my socre to 6.

---

### Official Review · Reviewer_3niS · 2022-07-26
**New NAS benchmark allowing independent searchable blocks**

**Rating:** 7
**Confidence:** 4
**Clarity:** The paper is well written and present…

**Strengths:**

Having a diverse set of search spaces is crucial for a proper assessment of NAS methods, and the macro search space provided by BLOX will be useful in this respect, especially given that most prior work focus on cell-based search spaces. The related queryable dataset will also allow fast evaluations of NAS methods and can therefore help speed up NAS research.

**Weaknesses:**

Despite the fact that “The search space is designed to be compatible with all NAS methods, including differentiable architecture search” as mentioned at the beginning of section 2, no differentiable NAS methods are applied to the search space. This would be useful to support the claim “1) Conventional NAS achieves worse results than standard blockwise (FT200) when a good teacher is used” made in section 3.3.

The fact that the Pareto-front of models with different blocks dominates that of models with uniform blocks, as highlighted towards the end of section 2 (highlight 1), is expected given the three orders of magnitude larger number of models with different blocks (45^3-45) compared to models with similar blocks (45) in the BLOX search space. Rather than comparing different and uniform-block architectures within the same search space, in order to motivate the study further, it would be useful to compare BLOX with a cell-based search space of roughly the same size (see also additional feedback below).

**Additional Feedback:**

Given that the BLOX search space is compatible with differentiable NAS methods as well, it would be useful to apply at least one well-known differentiable NAS method (preferably other than DARTS) to support (or disprove) the claim made in section 3.3 that “1) Conventional NAS achieves worse results than standard blockwise (FT200) when a good teacher is used”

If feasible, to motivate the creation of the BLOX search space further, it would be useful to include in figure 4 the Pareto-front of a cell-based search space (or perhaps sampled subsets of existing cell-based benchmarks, such as NAS-Bench-101 etc.) with roughly the same number of architectures and roughly the same range of number of parameters / flops.

**Correctness:**

I cannot identify any flaws in the paper, including the proposed search space, the construction of the dataset, and the performed experiments.

**Documentation:**

Sufficient details on documentation has been provided in the supplementary material section F, including data collection, maintenance, intended use etc., and the link to the repository including the source code has been made available.

**Ethics:**

There are no ethical concerns that warrant further discussion or review.

**Relation To Prior Work:**

Relation to prior work is discussed briefly in the introduction.

**Summary And Contributions:**

The Authors have introduced a new search space for NAS which allows for different cell architectures at each block, contrary to cell-based search spaces where a single cell architecture is repeated throughout the network.

The benchmark (referred to as BLOX) collects in particular information regarding the training and validation of all architectures on CIFAR-100 into a dataset, which can be queried through an API.

They further evaluate several conventional NAS algorithms, together with two recent block-wise NAS algorithms in the common setting provided by BLOX, and use the benchmark to analyze and address a number of questions regarding different components of block-wise NAS methods.

---

> ### Author Response · Authors · 2022-08-22
> **Thank you for the suggestions**
>
> We sincerely thank the reviewer for the valuable comments and positive scores.
>
> * No differentiable NAS methods are applied to the search space
>   * Thank you for your valuable suggestions. Please refer to **Section S3** for our response related to differentiable NAS methods. We have run DARTS-PT on Blox and have **added the results to Section 3.3 and Table 2**.
>
> * Compare BLOX with a cell-based search space of roughly the same size
>   * We have **added Figure 22** which plots Blox (macro search space, 91125 models) with NAS-Bench-201 (cell-based search space, 15625 models) as both have been trained on CIFAR-100. We also include NATS-Bench-SSS (32768 models). Please refer to **Section S1** for more details.
>
> We hope the responses above will resolve your concerns. If you have any more questions, please let us know and we would be happy to continue the discussion.

---

> > ### Comment · Reviewer_3niS · 2022-08-28
> > **Thanks for the updates**
> >
> >
> > Thank you for the updates and additional experiments. I appreciate the efforts made by the authors and would like to see the paper published.

---

### Official Review · Reviewer_L54G · 2022-07-27
**Benchmark for macro NAS**

**Rating:** 6
**Confidence:** 3

**Strengths:**

- The proposed benchmark focuses on macro NAS, which is a relatively new and less-studied field compared to cell-based NAS. The proposed search space of block operations is expansive compared to that of NAS-Bench-Macro. It contains more operations (up to 45) and includes the commonly seen ones such as residual blocks and inverted bottleneck blocks.

- The paper includes a detailed study on two blockwise NAS methods which have not appeared in previous NAS benchmarks.

- The analysis in the paper provides two valuable insights. First, the Pareto-frontier of searched models of macro NAS dominates that of models of micro NAS. This suggests that a macro search space contains higher performing models, thus pointing out a promising direction for future NAS research, i.e., to develop efficient methods that search for macro structures and allow heterogeneity between network blocks. Second, for blockwise NAS evaluation, the paper considers two novel settings—distillation and fine-tuning—in addition to the standard training setting. Performance results and the analysis provided can be helpful for the development of training methodology for future algorithms.

- Full experiment details, hyperparameters, and code are provided for reproducibility.


**Weaknesses:**

- The search space design has limited novelty and capacity. The set of block options trivial expands that of NAS-Bench-Macro. Moreover, as BLOX contains more block options, it only evaluates architecture with 3 stages to reduce computational cost (note that NAS-Bench-Macro search space has 8 stages). This design choice limits the depth and size of the candidate architectures, and consequently affects the learning capacity of the searched model. Nowadays, commonly used CNNs for image classification (even the simplest ones such as ResNet50) are much larger than 3-stage CNNs. This hinders the practical significance of the benchmark as few ML developers will actually use a network model that has a similar scale to that evaluated in the benchmark for research or industry purposes. Also, it is unsure whether the evaluated networks can work for more complicated tasks beyond CIFAR-100, and the analysis based on the searched results also might not generalize to more complicated architectures and difficult learning problems.

- The paper only evaluates two blockwise NAS algorithms. There are many other NAS methods that can be considered "macro", i.e., the ones that generate networks in a global view rather than stacking cells. Examples are morphism-based methods as mentioned in section 4.1.4 of [1]. Do these methods also count towards macro NAS? If not, can you give a better and clearer definition of macro NAS?

- The comparison of macro NAS and cell-based NAS in Fig. 4 might be unfair. When constructing the “uniform block” search space, it seems that the authors consider models generated with the same block options and the three-stage structure so that the number of candidate networks is far less than that of the macro search space. However, micro NAS methods are often more computationally efficient and work for significantly larger search spaces. For a fair comparison, I think the “uniform block” search space should be constructed in a way that either the size of the “uniform block” and the “different block” search space is identical, or the computational cost of exploring the two search spaces are similar. Right now, I find the conclusions of section 2.3 less convincing.

[1] He, Xin et al. “AutoML: A Survey of the State-of-the-Art.” Knowl. Based Syst. 212 (2021): 106622.


**Additional Feedback:**

Which blockwise NAS algorithm is used to obtain the results in Table 2?

**Clarity:**

The paper is mostly well-written but is less accessible to researchers working outside of the NAS domain.


**Correctness:**

I’m unsure about the claims made in section 2.3. The evaluation and experimental protocols in later parts seem correct and reasonable.


**Documentation:**

Yes, the code is provided and the experiment details, including the search and evaluation hyperparameters, are provided.


**Ethics:**

There seem to be no immediate ethical concerns.


**Relation To Prior Work:**

The authors discuss several related works in Introduction and Table 1 and make it clear that benchmarks and studies on macro NAS are lacking.


**Summary And Contributions:**

For computational efficiency, most existing NAS algorithms search for cell-based architectures. However, this paper argues that the search space of NAS algorithms should be “macro”, i.e., by including the full network topology in the search space and allowing operations to differ in each block. To systematically evaluate the performance of NAS algorithms in macro search spaces, the paper proposes BLOX, a NAS benchmark that consists of 91k CNN models with better block diversity trained on CIFAR100. Compared to previous macro NAS benchmarks, the BLOX search space is larger as it contains more block options. The paper also evaluates two recent blockwise NAS algorithms on the proposed search space and provides a detailed analysis of their performance and the implications on the efficacy of different block signatures, accuracy predictors, and training strategies.

---

> ### Author Response · Authors · 2022-08-22
> **Thank you for the comments**
>
> Thank you for the constructive and insightful feedback.
>
> * Regarding ‘search space design has limited novelty and capacity’, please refer to **Section S2**.
> * Regarding “generalisation beyond CIFAR-100 dataset”, please refer to **Section S5**.
> * Regarding "definition of macro NAS", please refer to **Section S3** where we give a definition of macro NAS and give our view on morphism-based methods. We have also **revised Section 1 (highlighted in red)** to clarify these points.
> * Regarding the compariosn of macro NAS and cell-based NAS, please refer to **Section S1** for a detailed response. In the revised paper, we have **added Figure 22** to compare Blox to the other search spaces with the same order of magnitude in terms of the number of architectures, parameters and FLOPs.
>
> We hope the responses above will resolve your concerns, and would consider increasing the score. If you have any more questions, please let us know and we would be happy to continue the discussion.

---

> > ### Comment · Reviewer_L54G · 2022-08-26
> > **Thanks for the response**
> >
> > Thank the authors for their detailed response and additional content. Most of my concerns are addressed. I'll increase my score to 6.

---

### Official Review · Reviewer_9Xhg · 2022-07-28

**Rating:** 6
**Confidence:** 4
**Correctness:** The dataset is constructed in a sound…
**Clarity:** The paper is well-written and easy to…

**Strengths:**

1. Macro search space and benchmark for NAS with 91k unique architecture. The accuracy improvement with the different blocks is remarkable according to Figure 4.
2. The analysis includes different NAS algorithms on Blox in terms of block signatures, accuracy predictors, and training methodologies. These factors include 2 existing work named DONNA and HANT.


**Weaknesses:**

1. The training process is extremely complex with three stage settings including Normal setting, Distillation and Fine-tuning according to Figure 6.

2. Also, with the results provided in Figure 7, the good or bad teacher results in different distilled and fine-tuned performance and has a relatively low spearman with accuracy when trained from scratch. Besides, the number of distilled epochs also affects the spearman correlation.
(1) Is it hard to choose a teacher? Or simply use the best teacher? How to choose the best teacher?
(2) The rank of accuracy distilled and fine-tuned by the bad teacher of all these models is not highly correlated with the rank of the good teacher as shown in Figure 10(b).
(3) which choice of distilled epoch makes more sense?

3. With these problems, is it hard to apply this pipeline for other researchers to use your Blockwise Search Space to perform architecture search in other network architectures? The complicated training pipeline and factors affected by teachers.
(1) what's the most important items?
(2) does there exist a more simplified procedure?

4. Can we use the Blox dataset to evaluate other commonly seen NAS algorithms besides DONNA and HANT?


**Additional Feedback:**

Please check the Weakness part.

**Documentation:**

Not very detailed documentation and promise to stay maintenance.

**Ethics:**

I did not see any ethical concerns.

**Relation To Prior Work:**

The difference from the previous contributions is clearly discussed.

**Summary And Contributions:**

This paper proposes a macro search space, which allows blocks in a model to be different to promote performance. To provide a systematic study of the performance of NAS algorithms in a macro search space named Blox – a benchmark that consists of 91k unique models trained on the CIFAR-100 dataset. The dataset also includes other hardware latency information.

---

> ### Author Response · Authors · 2022-08-22
> **Thank you for the comments**
>
> We sincerely thank the reviewer for the comments.
>
> * Regarding point 1, 2, 3 on the training process, particularly about choosing a teacher, choice of distillation epoch and the most important item in blockwise search on macro NAS. Please refer to **Section S6** in the summary for detailed answers.
>
> * Regarding point 4, please refer to **Section S3** on how we have used Blox to evaluate various common NAS methods (BRP-NAS, Regularised evolution, Hyperband, Random, Q-Learning, REINFORCE and DARTS-PT).
>
> We hope the responses above will resolve your concerns, and would consider increasing the score. If you have any more questions, please let us know and we would be happy to continue the discussion.

---

### Author Response · Authors · 2022-08-22
**Summary of responses**

We thank the reviewer for their constructive and insightful feedback. Please see the responses which are summarised into 7 sections below. For each individual reviewers’ comments, we also direct the response to the corresponding sections in the summary below in order to provide more concise answers in one place since many reviewers ask similar questions. We have also updated the paper with the changes highlighted in red colour. We hope the responses will resolve any concerns the reviewer might have, and we hope that in light of this the reviewers would consider increasing their score.

---

> ### Author Response · Authors · 2022-08-22
> **S7. Enable study of distilling the teacher on the dataset**
>
> Reviewer pU8w – this study is not enabled in the current setting.
> * Blox currently provides tabular results of training-from-scratch to enable systematic study of conventional NAS algorithms on emerging macro search spaces. Such results are also beneficial for studying blockwise algorithms (even though it does not directly enable their fast evaluation) because it allows better control of parameters of experiments (e.g. choosing "good teacher vs. bad teacher"), and enables comparison of the accuracy of the same models trained using different approaches.
> * Furthermore, blockwise NAS algorithms require selecting a teacher to perform blockwise distillation and fine-tuning. The performance is algorithm-dependent, thus we cannot provide a single dataset to enable fast evaluation. Also, any model in the search space can be a teacher, providing one dataset for each teacher would be beyond our capability at this stage.
> * We have **revised Section 2.2 (highlighted in red)** to clarify these points.

---

> ### Author Response · Authors · 2022-08-22
> **S6. Training process**
>
> Reviewer 9Xhg
>
> – The training process is complex according to Figure 6.
> * The training process consists of three main stages: 1) Blockwise distillation, 2) Search, 3) Fine-tuning. In the final revision, we make sure everything is as simple and clear as possible.
>
> – How to choose a teacher / the best teacher.
> * Regarding choosing a good teacher, we explore this in Section 3.2 – Q6 which leads to our proposed iterative approach that has significantly improved the model accuracy without knowing a good teacher in advance (see results in Figure 12, Section D3 and Table 5).
>
> – The ranking of models distilled by bad teachers is not highly correlated with that of the good teacher.
> * Figure 10b shows that correlation exists between the ranking of models distilled by bad teachers and that of the good teachers. This explains why the iterative approach works well in this setting. In other words, to make it very clear, our experiments show that when using a bad teacher we can in fact find good architectures (to some extent) and the main challenge is rather related to the fact that we cannot train them to their full potential (because the teacher performs poorly).
>
> – Which choice of distillation epoch makes more sense?
> * Regarding the choice of distillation epoch, we have the results summarised in Section D.2 – Q8. It shows that distillation for only 1 epoch leads to worse predictors, the predictor performance improves with the number of block distillation epochs, resulting in a typical cost vs. quality trade-off.
>
> – Is it hard to apply this pipeline for other researchers to Blockwise Search Space to perform architecture search in other network architectures? What's the most important item? Does there exist a more simplified procedure?
> * For conventional NAS algorithms (multi-trial, query-based), researchers can reuse their existing implementations and easily query our tabular results in the Blox macro search space, using the API provided in the source code. Other methods, including blockwise algorithms, alter training of models in an algorithm-dependent way thus making tabular results not directly applicable. Still, after a particular architecture is identified, checking its standalone performance (and possibly comparing to the results obtained by the altered training process, e.g. like in Figure 7) is easy. We include code to construct and train models in our codebase to make it easier to use our search space with methods that rely on changing how models are trained.
> * Regarding the most important item of blockwise NAS methods (e.g. the most important items), we asked a series of questions in Section 3, specifically, fine-tuning vs training-from-scratch (Q1-2), settings of predictors (Q3-4, Q7), fine-tuning epochs (Q5), distillation epochs (Q8) and selection of teachers (Q6). In the final revision, we will make sure everything is as simple and clear as possible.
>
> Reviewer dMTQ
>
> – Q1-Q6 are interesting but a little bit hard to follow. Why do you ask these questions and how are they connected, and please also highlight the concrete answer (bold) for each question.
> * We asked a series of questions in order to isolate relevant behaviour of the studied algorithms – 1) fine-tuning (improvement brought by fine-tuning, the impact of teacher, correlation between fine-tuned models and conventionally trained models); 2) predictor (How do block signature and end-to-end predictor affect the performance of blockwise NAS methods?); 3) search efficiency (Can we fine-tune end-to-end model and distil blocks with few epochs? Can we fine-tune with a bad teacher?).
> * The answers lead to a consistent conclusion – performance is improved on blockwise methods over conventional algorithms, as presented in Section 3.4 (Figure 16 and Table 2). In summary, 1) when a good teacher is used, blockwise NAS achieves better results (i.e. more accurate model and lower search cost) than conventional NAS ; 2) when a bad teacher is used, we proposed a simple iterative strategy which allows us to again dominate conventional NAS. 3) efficiency of blockwise NAS is improved by utilising reduced fine-tuning proxy followed by full fine-tuning, which is our contribution stemming from questions Q 1-6.
> * In the paper revision, we have **highlighted the answers in bold** to make them clear.

---

> ### Author Response · Authors · 2022-08-22
> **S5. Only evaluated on CIFAR-100**
>
> Reviewer pU8w – CIFAR-10 is the common denominator for NAS benchmark datasets.
> * CIFAR-100 is a harder task than CIFAR-10 so the differences between architectures are magnified, which we thought would make the benchmark more interesting. Also, there is a lack of benchmarks on CIFAR-100, while CIFAR-10 is much more well-studied in this context. * Furthermore, existing NAS methods have usually been tested on different datasets, including CIFAR-10, and it is advised to test new methods on more than one benchmark, therefore, our new benchmark on CIFAR-100 would not make it less usable for NAS researchers.
> While we agree that it is desired to include more datasets in general, unfortunately due to time and computing constraints, training all 91,125 models on more datasets is beyond our capability. In the released code base, we will include the training scripts which allow extending the benchmark to support other datasets.
>
> Reviewer L54G – Generalisation beyond CIFAR-100 and more complicated tasks / learning problems.
> * Regarding generalisation of our analysis, we agree that it should be approached carefully. However, this is a more generic criticism of NAS benchmarks not specific to our work - it is computationally intractable to produce a NAS benchmark that would study problems and models significantly larger than those typical for CIFAR-10(0) scale. A common remedy for that is to transfer only selected architectures (e.g. 5-10) to a larger task and show some desired properties but in our opinion statistical significance of such experiments is usually questionable and therefore they have secondary meaning. Overall, we do not think our BLOX benchmark is particularly flawed in that regard compared to the existing benchmarks. Therefore, if the reviewer agrees that NAS benchmarks in general are beneficial to the community, we hope that they would agree that our BLOX is only going to make the situation better by making the landscape of NAS benchmarks more comprehensive.

---

> ### Author Response · Authors · 2022-08-22
> **S4. Quality of the search space**
>
> Reviewer pU8w – the best accuracy is 76.6%, lower than 80% reported in P-DARTS.
> * The goal of Blox is to provide a new macro search space and benchmark to enable study of emerging blockwise NAS algorithms. There is increasingly more NAS research focusing on the macro search spaces (e.g. NAS-Bench-Macro, DONNA, HANT, DNA) that allow individual search for each block in a DNN. It serves a similar purpose as other benchmarks (e.g. NAS-Bench-101, NAS-Bench-201, NAS-Bench-Macro) which do not beat the SOTA models.
>
> Reviewer dMTQ – clarify the reason or insight of the search space. Why and what are the benefits?
> * Regarding the insight of the macro search space, a macro search space enables layer diversity (Figure 1), and provides a trade-off between achievable results and amount of configurations available (Figure 4). However, the macro search space size grows exponentially with the number of blocks, posing a challenge to the existing search algorithms. On the other hand, blockwise search algorithms are emerging (particularly DONNA and HANT that are studied in our paper), however, different methods are not comparable to each other due to different training procedures and different search spaces. This paper presents the first large-scale benchmark on macro search space, which enables efficient ways to study NAS in this challenging setting.

---

> ### Author Response · Authors · 2022-08-22
> **S3. Applying other NAS methods, particularly differentiable NAS method**
>
> Reviewer 3niS – apply at least one well-known differentiable NAS method (preferably other than DARTS) to support (or disprove) the claim made in section 3.3 that “1) Conventional NAS achieves worse results than standard blockwise (FT200) when a good teacher is used”.
>
> Reviewer pU8W – excluded weight-sharing (one-shot) methods that are state-of-the-art.
>
> * We have run DARTS-PT on Blox and have **added the results to Section 3.3 and Table 2**.
> 1. Regularized Evolution - 76.10
> 2. BRP - 76.40
> 3. DARTS-PT - 74.52
>
> Reviewer pU8W – only a limited number of NAS methods are evaluated.
>
> Reviewer 9Xhg – can we use the dataset to evaluate other common NAS algorithms?
> * We have already used Blox to evaluate the performance of various common NAS methods (BRP-NAS, Regularised evolution, Hyperband, Random, Q-Learning, REINFORCE) in Table 2, Figure 5, Figure 20 and Figure 21. The efficiency of blockwise NAS vs. conventional NAS are studied in Section 3.3 and Figure 20.
> In the lastest paper revision, we have also **added the results of DARTS-PT in Table 2**.
>
> Reviewer L54G – give a better and clearer definition of macro NAS, does morphism-based methods count?
> * We define macro NAS as a NAS problem where different stages of a model are allowed to have different structures (Section 1 and Figure 1, Section 2.1 and Figure 2). Please note that this definition is unrelated to what algorithm is used - as we show in our paper we can use many conventional algorithms. On the other hand, blockwise NAS methods, which are the main focus of our paper as they have recently gained popularity, are simply an example of NAS algorithms that were specifically designed with macro NAS in mind. As described in Section 3 and Figure 6, we define blockwise NAS methods as methods that consist of 3 stages: 1) Blockwise distillation; 2) Search, and 3) Fine-tuning. To clarify, we did not intend to say that blockwise NAS algorithms are the only ones that are expected to work well for macro NAS (that’s why we include comparison to conventional algorithms in our paper). To the best of our knowledge, morphims-based methods are indeed well-suited for macro NAS but admittedly we do not include any method like that in our comparison at the moment. If the reviewer thinks it would make the paper noticeably stronger, we might be able to include a representative method by the camera ready deadline.
> * We have **revised Section 1 (highlighted in red)** to clarify these points.

---

> ### Author Response · Authors · 2022-08-22
> **S2. Novelty and capacity**
>
> Reviewer L54G – limited novelty and capacity, NAS-Bench-Macro has 8 stages. 3 stages limits the depth and size, hinders the practical significance of the benchmark.
>
> * We disagree with the statement that our choice to use 3 stages limits the depth and size of networks. Specifically, please note that although models from the blox search space are broken down into fewer parts compared to NAS-Bench-Macro, we consider a larger number of more diverse replacements. In terms of individual linear operations (e.g., a single convolution) the shallowest block out of the 45 candidates in our search space contains only 4 layers while the deepest 36. This means that the whole network can be as shallow as 12 layers or as deep as >100 (excluding fixed parts). Actually, this range is larger than that in NAS-Bench-Macro where networks’ depth, in terms of searchable linear operations, varies from 0 to 24.
> * In fact, the number of stages in a search space does not directly influence depth of networks but rather simply changes granularity of choices to which a searching algorithm is exposed, and the depth is controlled by the available choices themselves rather than their granularity. In that regard our design of blox follows a complementary approach where we use lower granularity of blocks with more diverse replacements, while NAS-Bench-Macro focuses on the opposite direction where higher granularity of blocks also results in lower diversity of candidates (for example, because searchable stages in NB-Macro only contain a single operation it does not contain any choices regarding connectivity, while in blox we use 2 operations per searchable stage thus introducing another degree of freedom related to connections between them).
> * We hope that the above makes it clear that our blox design is complementary to NB-Macro with some important qualitative differences (on top of simply being larger), which was our goal. Consequently, we hope that it also makes it clear why it is not simply a trivial extension of NB-Macro and that its practical significance is notable - having NAS benchmarks that explore different design choices increases our opportunities to study NAS algorithms in different situations and better understand their behaviour.
> * We have **revised Section 2.3 (highlighted in red)** to clarify these points.

---

> ### Author Response · Authors · 2022-08-22
> **S1. Comparison with cell-based search space**
>
> Reviewer 3niS, Reviewer L54G – include in figure 4 the Pareto-front of a cell-based search space (or perhaps sampled subsets of existing cell-based benchmarks, such as NAS-Bench-101 etc.) with roughly the same number of architectures and roughly the same range of number of parameters / flops.
> * In the revised paper, we have **added Figure 22** which plots Blox (macro search space, 91125 models) with NAS-Bench-201 (cell-based search space, 15625 models) as both have been trained on CIFAR-100. We also include NATS-Bench-SSS (32768 models) which is based on NAS-Bench-201 but scales the architecture size rather than topology. This figure compares Blox to the other search spaces with the same order of magnitude in terms of the number of architectures, parameters and FLOPs.
> * Please note that the purpose of Figure 4 is to show the trade-off between achievable results and the amount of configurations available. Every cell-based search space can be turned into a much larger macro search space (and lead to a much higher exploration cost), and the achievable accuracy would likely improve (worst case scenario they would stay the same, but won’t be worse since cell-based is a subset of macro).

---

### Author Response · Authors · 2022-08-25
**Feedback on responses**

Dear reviewers and AC,

We have now addressed the suggestions and concerns mentioned by the reviewers. Thank you very much for these comments which have substantially improved our work.

Our responses and changes in the new version of our paper are listed in S1 to S7 below (Summary of responses).

We are happy to address any new follow-ups or concerns before the response period ends on August 29.

Many thanks.

---

### Comment · Area_Chair_P9bH · 2022-08-26
**Urgent need for discussion**

Dear reviewers,

Thanks everyone for your reviews. The authors have replied at length, but so far none of you have replied to them or to each others’ reviews. This is an integral part of the OpenReview process, so I would like to ask you to read each others’ reviews and the authors’ rebuttal, and at least post a message saying that you did so and it doesn’t change your rating (or, of course, additional points).

In forming my recommendation I will weigh the opinion of responsive reviewers higher than that of unresponsive reviewers, since it is based on more information.

Best,
AC

---

### Public Comment · ~prabhant_singh1 · 2022-11-07
**Code still missing.**

Hi, I noticed that the code for this benchmark is still missing. Can authors confirm if it is still under processing as the GitHub repo is still empty.

---

### Meta-Review · Area_Chair_P9bH · 2022-09-15

**Recommendation:** Accept
**Confidence:** 3

**Metareview:**

This paper introduces a new NAS benchmark with 95k architectures evaluated on a single dataset.
Criticisms were quite diverse, including the novelty and design of the search space, the use cell-based baselines with much smaller search spaces, the limited number of algorithms being benchmarked, and the limitation to a single dataset, CIFAR-100. However, most of the criticisms were addressed during the rebuttal, leading to several reviewers increase their scores.
Overall, all reviewers are in favour of acceptance, some of them clearly so. I therefore recommend acceptance as a poster.

---

### Decision · Program_Chairs · 2022-09-16

Accept